# In situ observations of an active $MoS_2$ model hydrodesulfurization catalyst

Rik V. Mom [1], Jaap N. Louwen[2], Joost W.M. Frenken[1,3] & Irene M.N. Groot [1,4]

The hydrodesulfurization process is one of the cornerstones of the chemical industry, removing harmful sulfur from oil to produce clean hydrocarbons. The reaction is catalyzed by the edges of $MoS_2$ nanoislands and is operated in hydrogen-oil mixtures at 5–160 bar and 260–380 °C. Until now, it has remained unclear how these harsh conditions affect the structure of the catalyst. Using a special-purpose high-pressure scanning tunneling microscope, we provide direct observations of an active $MoS_2$ model catalyst under reaction conditions. We show that the active edge sites adapt their sulfur, hydrogen, and hydrocarbon coverages depending on the gas environment. By comparing these observations to density functional theory calculations, we propose that the dominant edge structure during the desulfurization of $CH_3SH$ contains a mixture of adsorbed sulfur and $CH_3SH$.

[1] Huygens-Kamerlingh Onnes Laboratory, Leiden University, Niels Bohrweg 2, 2333 CA Leiden, The Netherlands. [2] Analytical Research and Quality Department, Albemarle Corporation, Nieuwendammerkade 1-3, 1022 AB Amsterdam, The Netherlands. [3] Advanced Research Center for Nanolithography, Science Park 110, 1098 XG Amsterdam, The Netherlands. [4] Leiden Institute of Chemistry, Leiden University, Einsteinweg 55, 2333 CC Leiden, The Netherlands. Correspondence and requests for materials should be addressed to R.V.M. (email: mom@physics.leidenuniv.nl)

The hydrodesulfurization (HDS) process is used to remove environmentally harmful sulfur from ~2500 million tons of oil annually[1]. Thus, it is an essential step in the production of clean fuels. To accomplish the removal of sulfur, the oil is mixed with hydrogen at a pressure between 5 bar and 160 bar, with the temperature between 260 °C and 380 °C, producing $H_2S$ and clean hydrocarbons[2].

$MoS_2$-based catalysts are widely used to drive the HDS reaction owing to their high activity, stability, and low cost[2]. While research into these catalysts started as early as the 1920's[3], the atomic-scale mechanism of the reaction is still under debate. Ex situ microscopy data showed that $MoS_2$ is present as nanoislands that consist of one or more layers of an S–Mo–S sandwich[4–10]. The edges of the islands serve as active sites during catalysis[11–14]. Hence, mechanistic understanding of the HDS process requires detailed knowledge of the properties of the edge sites.

In most cases, the $MoS_2$ islands exhibit a high degree of crystallinity, resulting in island shapes close to the thermodynamically favored truncated triangle[4–8,15,16]. Consequently, two types of edge termination are mainly observed, usually referred to as the Mo edge and the S edge, with the Mo edge being dominant. Ab initio thermodynamics calculations predict that the sulfur coverage on both edge structures depends on the gas environment[15,17–19]. The essential element is the balance between the chemical potentials of hydrogen and sulfur atoms in the gas phase. In a sulfur-rich feed, two S edge atoms per Mo edge atom (100% coverage) are predicted. Excess hydrogen can lower this number, in the limit of pure $H_2$ even to zero (0% coverage).

Experimental studies in several gas environments confirmed that the edge sulfur coverage varies, depending on the balance between hydrogen-containing and sulfur-containing species in the feed[15,20,21]. However, it remains extremely challenging to study a minority species such as edge atoms under *operando* conditions. As a result, a conclusive determination of the active site structure during HDS has so far not been achieved.

Here, we present direct observations of the active $MoS_2$ edge structure under reaction conditions. Using a special-purpose high-pressure scanning tunneling microscope (STM), we have obtained atomically resolved images evidencing a mixed hydrocarbon-sulfur edge structure during the desulfurization of $CH_3SH$ on a model catalyst. We explain the observations by comparing the STM images with density functional theory (DFT) calculations, also taking into account the role of the reaction kinetics in determining the active site structure.

## Results

**Instrument development and model system.** The combination of high pressures of corrosive gases and high temperature provides a challenging environment for STM experiments. To meet this challenge, we have used the ReactorSTM previously developed in our group[22]. At the heart of this system is a 0.5 ml flow reactor containing the STM tip. The reactor walls are defined by a cap placed inside the scan piezo, the catalyst sample, the STM body, and polymer seals in between these elements. Gas capillaries drilled in the STM body provide a connection to a gas supply system that controls the composition, flow, and pressure of the gases in the flow reactor.

To mimic realistic industrial conditions, the pressure can be raised up to 1 bar, while the sample is heated up to 300 °C using a filament located on the backside of the sample. A key element in the design is that the scan piezo is not in contact with the gases in the reactor. This geometry, in combination with a careful choice of chemically resistant materials (see "Methods" section), allows for the use of highly corrosive gases such as $H_2S$ at elevated temperatures. The use of PtIr tips prevents tip degradation during the measurements, albeit that frequent changes in imaging quality cannot be avoided. To combine the high-pressure experiments with more traditional ultrahigh vacuum (UHV) surface preparation and characterization techniques, the flow reactor can be opened and closed inside a UHV system, which contains among others an ion gun, an evaporation source, and an X-ray photoelectron spectroscopy apparatus.

As a first step, we established a suitable model catalyst; one that is conductive to allow for STM measurements and stable under HDS conditions. Following the successful recipe of the Århus group[4,15], we synthesized $MoS_2$ particles on a Au(111) substrate (see "Methods" section), yielding crystalline islands with a predominantly triangular shape (see Fig. 1a). These islands were shown to nearly exclusively expose the Mo edge[15], which we will focus on hereafter.

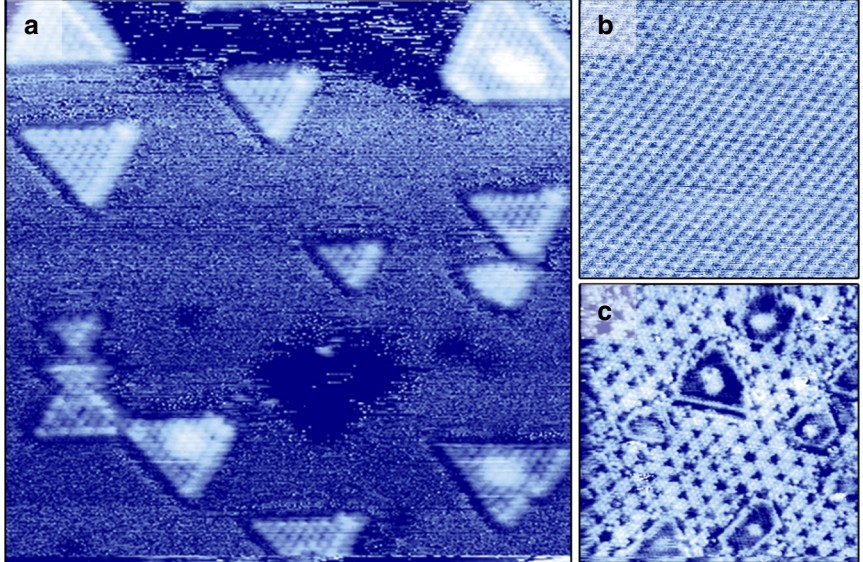

**Fig. 1** STM images of a $MoS_2$/Au(111) model catalyst and its stability under desulfurization conditions. **a** Catalyst after preparation in UHV. 16 × 16 nm², $U_{bias} = -0.3$ V, $I_t = 320$ pA. **b** Clean Au surface imaged in 1 bar $CH_3SH$ at 250 °C, showing the (1 × 1) Au lattice. 6 × 6 nm², $U_{bias} = -0.3$ V, $I_t = 550$ pA. **c** $MoS_2$/Au(111) after 1 day in 1 bar of a 1:9 $CH_3SH/H_2$ mixture, showing a sulfur overlayer on the Au(111) substrate. 20 × 20 nm², $U_{bias} = -0.3$ V, $I_t = 645$ pA

To allow for unambiguous identification of edge structures in STM during HDS, we chose the simple CH$_3$SH molecule as our organosulfur compound to be desulfurized. Naturally, a single organosulfur compound can never fully represent the complex mixture in oil. However, mercaptans such as CH$_3$SH constitute a major component in crude oil[23]. Our applied temperature and pressure (250 °C, 1 bar) are close to the typical hydrodesulfurization conditions applied for the light naphtha fraction (260–380 °C, 5–10 bar)[2] and are sufficient to achieve catalytic turnover[24].

The stability of the model catalyst during HDS depends on the chosen conditions. Like all thiols, CH$_3$SH readily adsorbs on gold surfaces[25]. However, at the temperature of our catalytic experiments (250 °C), only the (1 × 1) Au lattice is imaged, even in 1 bar CH$_3$SH (see Fig. 1b). Nonetheless, the absence of the herringbone reconstruction observed on clean Au(111)[26] indicates that some (dissociated) CH$_3$S is present on the surface. Decomposition of CH$_3$S or H$_2$S leads to the formation of a sulfur layer over time (see Fig. 1c), with a structure resembling that observed by Lay et al.[27]. The formation of the sulfur overlayer occurred independent on whether MoS$_2$ particles were present on the Au(111) substrate or not. As long as H$_2$S is not added to the reactor feed, the overlayer formation requires hours, making it too slow to interfere with the HDS catalysis on the MoS$_2$ particles through sulfur spillover. However, the encapsulation of the MoS$_2$ particles by the sulfur overlayer on the Au substrate, depicted in Fig. 1c, could limit the accessibility of the active edge sites. To prevent this, we restricted the duration of our HDS experiments to a few hours, after which a fresh model catalyst was prepared.

**Experimental observations on MoS$_2$ edge structures.** As a next step, we characterized the appearance of the fully sulfided MoS$_2$

edge structure obtained after preparation of the MoS$_2$ particles in $2 \times 10^{-6}$ mbar pure H$_2$S. Using DFT calculations, Lauritsen et al.[15] identified the resulting edge structure as the 100%S edge, which contains an S dimer on every Mo edge atom. Its appearance in STM images is characterized by a periodicity of one lattice spacing along the edge and by a registry shift of half a lattice spacing in the apparent position of the edge atoms with respect to those on the basal plane. Indeed, Fig. 2a shows that the apparent location of the edge atoms does not follow the registry of the basal plane atoms, as also confirmed by the non-differentiated image in Supplementary Fig. 3. The observed registry shift is smaller than the expected half lattice spacing, which was also observed in some cases in the literature[9,15,28]. We attribute this to asymmetry in the tip apex, which can lead to slight distortions in the observed edge structure.

Note from the 100% S ball model in Fig. 2a that the apparent position of the edge atoms in STM deviates from their geometrical position. This is a result of the fact that the electronic states around the Fermi level, which are probed by STM, are mostly located in between the edge atoms (see Supplementary Fig. 1a). To clearly separate the nomenclature for apparent and actual edge atom positions, we refer to the apparent positions of the edge atoms as edge protrusions hereafter. We should also point out that the STM images in Fig. 2 were differentiated to highlight the atomic contrast. In this display method, the metallic "bright" edge states that are usually observed at the edges of the MoS$_2$ particles are less apparent.

Our first step towards reaction conditions is to image the model catalyst in high H$_2$ pressure and at high temperatures. Ab initio thermodynamics calculations predict that these conditions should trigger a shape change in the particles to a more hexagonal shape (as opposed to the predominantly triangular shape of the as-prepared particles). However, we did not observe a decrease in the fraction of

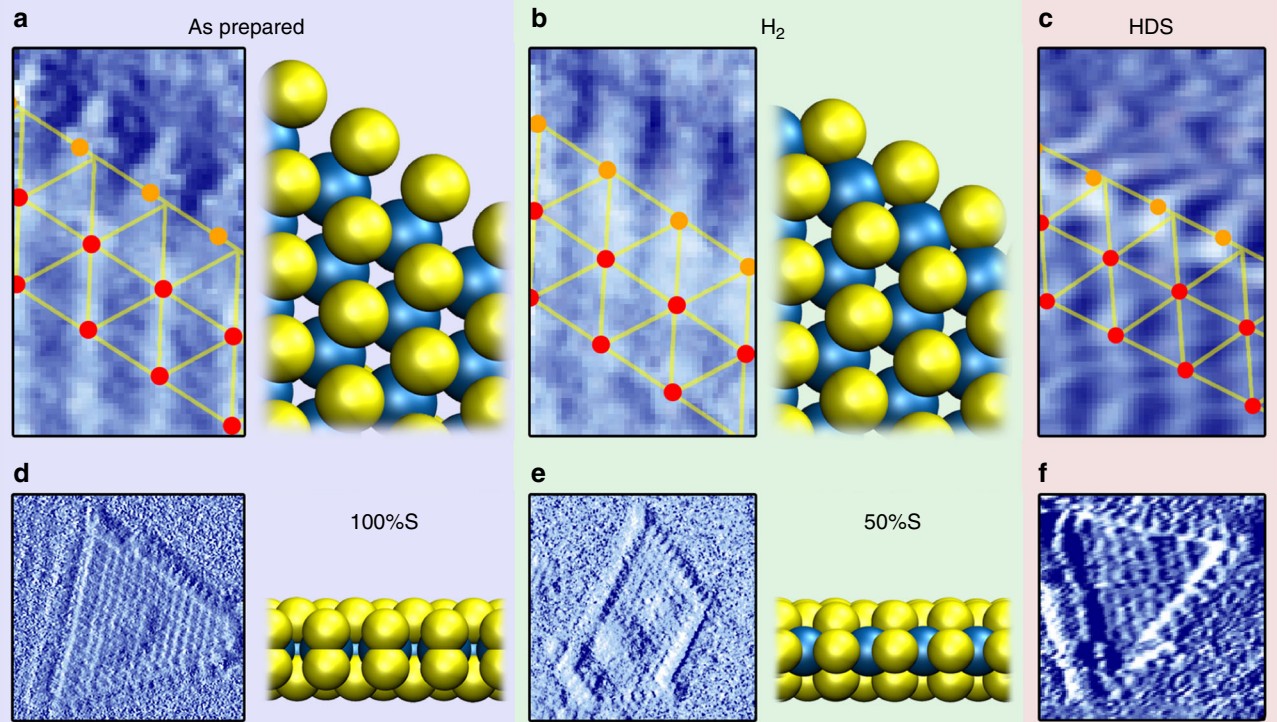

**Fig. 2** MoS$_2$ edge structure in various gas environments. The bottom panels (**d**, **e**, **f**) show the original images, which were differentiated (in the fast scanning direction) to highlight the atomic contrast. The top panels (**a**, **b**, **c**) depict the averaged edge unit cell obtained from the bottom panels (**d**, **e**, **f**). The ball models represent the 100%S and 50%S structures that could directly be identified for (**a**) and (**b**), respectively, from comparison to simulated STM images (see Supplementary Figs. 1 and 2). Blue: Mo, yellow: S. **a**, **d** Catalyst after preparation in $2 \times 10^{-6}$ mbar H$_2$S at 450 °C, imaged in UHV at room temperature. 6.6 × 6.6 nm$^2$, $U_{bias} = -0.3$ V, $I_t = 560$ pA. **b**, **e** Catalyst imaged in 1 bar H$_2$ at 50 °C. 6.6 × 6.6 nm$^2$, $U_{bias} = -0.3$ V, $I_t = 630$ pA. **c**, **f** Catalyst during the desulfurization of CH$_3$SH in 1 bar 1:9 CH$_3$SH/H$_2$ at 250 °C. 8 × 8 nm$^2$, $U_{bias} = -0.3$ V, $I_t = 400$ pA

triangular particles, which we attribute to kinetic limitations. Indeed, shape changes were experimentally only observed at higher temperatures[29]. On the more detailed scale however, we see that the appearance of the atomic structure of the edge sites does change (see Fig. 2b), indicating that the particle edges have been reduced. The registry shift of the edge protrusions with respect to the basal plane atoms that was apparent for the 100%S edge has been removed, while maintaining the periodicity along the edge of one lattice spacing. Note again that some tip asymmetry could not be avoided, resulting in somewhat asymmetrical image sharpness. To prevent misinterpretation, the conclusions from Fig. 2b were corroborated using additional data (see Supplementary Fig. 4). In all cases, we find that the edge protrusions are located precisely in registry (average 2% of a lattice spacing shift measured with respect to the basal plane registry). The structure in Fig. 2b was not dependent on temperature within the range from 50 °C to 250 °C probed here. It should be noted however that at elevated temperatures STM may probe a time-averaged structure, which could obscure diffusing S vacancies or adsorbed S and H atoms.

The edge structure without registry shift in Fig. 2b was also observed after deposition of Mo in $H_2S/H_2$ mixtures[15] or dimethylsulfide[30]. It was interpreted as the 50%S structure by comparison with simulated STM images. Our STM simulations corroborate this assignment, although hydrogen adsorption cannot be excluded (see Supplementary Figs. 1 and 2). In agreement with the observations on reduced $MoS_2$ edge structures in the literature, the bright metallic edge states are maintained (see Supplementary Fig. 7). We note that Bruix et al. did not find the registry shift between the 100%S and the 50%S edge structures for $MoS_2$/Au(111) in their STM simulations[21]. However, none of the structures described in their work matches the experimental observations of the reduced edge structure.

Having established our ability to observe changes in the $MoS_2$ edge structure under high-pressure, high-temperature conditions, we are ready to study our model catalyst in its active form during the desulfurization of $CH_3SH$. Fig. 2c shows the structure observed in a 1 bar 1:9 $CH_3SH/H_2$ mixture at 250 °C. Before imaging, the flow reactor was allowed to stabilize for more than

2 h to ensure a steady-state situation. Fig. 2c shows that the edge structure under hydrodesulfurization conditions has changed with respect to the structure in pure hydrogen (see also Supplementary Fig. 5 and see Supplementary Fig. 6 for the alignment procedure of the grid overlay). The observed registry shift of the edge protrusions is similar to that observed for the 100%S-covered edge in Fig. 2a: on average 20% of a lattice spacing out of registry in Fig. 2a and 18% in Fig. 2c and S5. Again, we note that the registry shift is asymmetric with respect to the lattice, which we attribute to asymmetry in the tip apex. Indeed, one can observe that in both Fig. 2a, c the resolution on the basal plane atoms is higher for the horizontal direction than for the vertical direction, which explains why the two edge structures show the same asymmetry. Based on the registry of the edge protrusions, one could assign the structure under HDS conditions to an (almost) 100%S-covered edge. An alternative explanation is the formation of $CH_3SH$ adsorption structures. Mo carbide formation can be excluded, since our low-temperature, sulfur-rich HDS environment is far away from the conditions for which $Mo_2C$ or $MoS_xC_y$ formation were observed[31–33].

**Theoretical modeling.** To enable an unambiguous assignment of the edge structure observed during the hydrodesulfurization of $CH_3SH$, we consider the thermodynamic and kinetic aspects of the catalytic process using DFT calculations. First, we assess the thermodynamic stability of various edge structures in a gas environment that only consists of $H_2$ and $H_2S$. Fig. 3 depicts the most stable edge structure for Au-supported $MoS_2$ as a function of $\Delta\mu_S$ and $\Delta\mu_H$. These quantities are directly related to the temperature, the $H_2S$ pressure, and the $H_2$ pressure through Eqs. 5 and 6 in the "Methods" section. The phase diagram in Fig. 3 corroborates the observation that the $MoS_2$ edge structure depends on the gas environment, showing large variations in both S and H coverage. A quantitative comparison with earlier work shows an agreement to within 0.15 eV for the relative stability of the 50%S and 100%S structures[15,18,19] (see Supplementary Table 1). Remarkably however, we find a preference for low-symmetry structures such as 38%S-x%H and 63%S-x%H over a

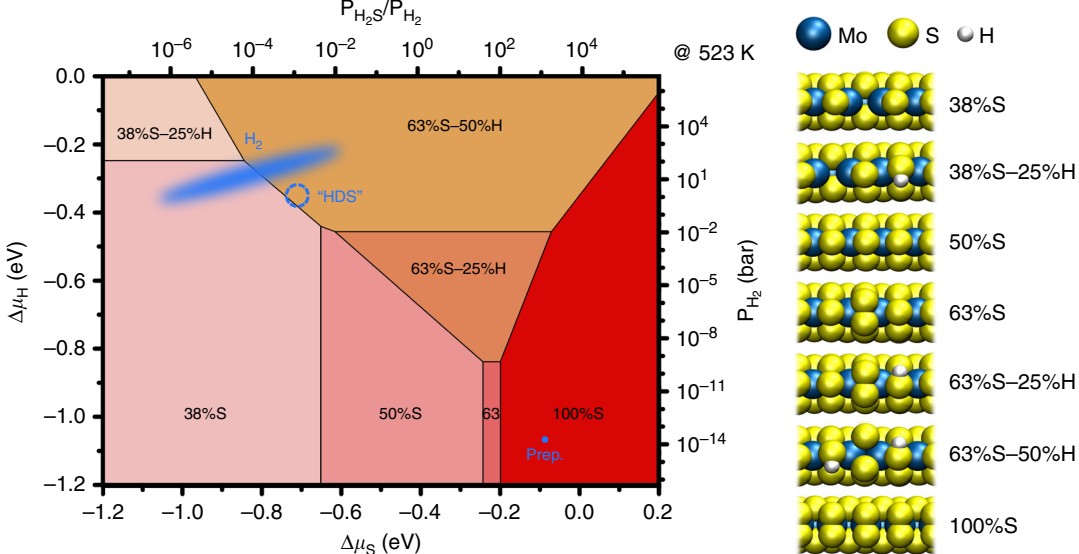

**Fig. 3** Ab initio thermodynamics phase diagram of the $MoS_2$ edge structures in $H_2/H_2S$ mixtures for Au-supported $MoS_2$. On the axes, $\Delta\mu_S$ and $\Delta\mu_H$ designate the entropic parts of the chemical potentials of S and H atoms in the gas phase, respectively. These are directly related to the temperature, the $H_2S$ pressure, and the $H_2$ pressure (see Methods section). The experimental conditions during catalyst preparation, in 1 bar hydrogen (assuming 1 ppm $H_2S$ contamination), and during HDS (naïve approximation) are indicated in blue. The ball models represent side views of the structures present in the phase diagram. Note that the edges are periodic in the left-right direction

wide range of conditions. These structures were not taken into account in earlier studies and therefore did not show up in the predicted phase diagrams (see Supplementary Note 3 for details). We have excluded that the stability of the low-symmetry structures is the result of the Au-MoS$_2$ interaction (see Supplementary Note 2), which generally favors a higher sulfur coverage, but does not seem to favor the low-symmetry structures in particular. However, we should point out that for a larger unit cell size, an even larger variety of structures may appear in the phase diagram.

The trends of the sulfur and hydrogen coverages in various gas environments can be understood based on the variation in adsorption strength per H or S atom of the various structures. Generally, one expects the adsorption strength per S or H atom to decrease at higher S or H coverage. Hence, higher-coverage structures require a higher chemical potential in order to form. This trend is indeed observed in the phase diagram. However, the 75%S and 88%S structures do not appear. In this coverage range, the registry of the S edge atoms changes with respect to the S atoms on the basal plane, leading to unstable structures with strained bonds. For the hydrogenated phases, 63%S-x%H shows a remarkably large range of stability. An explanation for this comes from the comparison of the 50%S and 63%S structures. For the 50%S case, hydrogen adsorption induces buckling of the edge S atoms (Supplementary Fig. 1), implying the presence of compressive stress. The buckling disappears upon the adsorption of a sulfur atom (yielding the 63%S-50%H structure, see Fig. 3), implying a stabilizing stress relief. For the small particles used in HDS, corner sites may provide similar stress relief. It is therefore not a priori clear whether the 63%S-x%H structures are similarly stable in such more realistic catalysts.

Using Eqs. 5 and 6 (see Methods section), we have placed the experimental conditions in the phase diagram of Au-supported MoS$_2$. For the freshly prepared particles, we have used an H$_2$S pressure of $2 \times 10^{-6}$ mbar H$_2$S and a temperature of 300 °C for the calculation, even though imaging was performed in vacuum at room temperature. We chose these conditions because the edge structure is not capable of changing in vacuum at temperatures below 300 °C[15,21]. As expected, the phase diagram indicates the observed 100%S structure to be the most stable under these conditions. For the reduction in 1 bar H$_2$, we assumed an H$_2$S contamination level of 1 ppm. Depending on the temperature, the phase diagram indicates an edge coverage of 38%S to 63%S, with hydrogen adsorption on most structures. Again, this is in good agreement with the time-averaged 50%S-X%H structure observed with STM.

To represent the hydrodesulfurization experiment in the phase diagram, we need to assume that the gas environment can be described solely in terms of H$_2$S and H$_2$ chemical potentials. This would be the case if the adsorption of hydrocarbons, e.g., CH$_3$SH, is ignored and the overall HDS reaction is either completely equilibrated or slow with respect to the reactions of H$_2$ and H$_2$S with the MoS$_2$ edges. From mass spectroscopic product analysis, we know that in our case the conversion of CH$_3$SH is low due to the

extremely low number of active sites on our planar model catalyst. Even at an extremely high turnover frequency of 1000 s$^{-1}$ per site, only ~1 mbar H$_2$S would be generated. If we assume this upper limit and ignore CH$_3$SH adsorption, the chemical potential of sulfur in our HDS experiment is determined by the temperature (250 °C), the H$_2$S pressure (~0.001 bar) and the H$_2$ pressure (0.9 bar), as indicated in Fig. 3. This would lead to a 63%S-50%H structure, which has a slightly higher S coverage than the 50%S-50%H structure predicted in earlier studies for these conditions[15,18]. It should be clear however that the 100%S structure is not a likely candidate for the structure we observe in STM under reaction conditions.

To investigate the possibility of CH$_3$SH adsorption structures, we calculated the CH$_3$SH adsorption energy for several edge S and H coverages on Au-supported MoS$_2$ (see Supplementary Note 4). While CH$_3$SH can bind in all cases, only the 38%S-25%CH$_3$SH structure in Fig. 4 is thermodynamically more stable ($\Delta G_{form} = -0.08$ eV) than the clean 63%S-50%H structure under our reaction conditions (P$_{CH3SH}$ = 0.1 bar, P$_{H2S}$ = ~0.001 bar, P$_{H2}$ = 0.9 bar, 250 °C). Hence, if H$_2$, H$_2$S, and CH$_3$SH would equilibrate with our model catalyst, i.e., if the HDS reaction would be slow enough not to affect this thermodynamic equilibrium, the 38%S-25%CH$_3$SH structure should prevail. For H$_2$S pressures lower than the upper limit of 1 mbar, the preference for the 38%S-25%CH$_3$SH structure is further increased. Fig. 4b shows an STM simulation of the 38%S-25%CH$_3$SH structure. The irregular appearance of the edge structure will be time-averaged in the STM images at 250 °C, because of the fast adsorption/desorption kinetics of CH$_3$SH (free energy barriers of 0.51 eV and 0.87 eV, respectively). In Fig. 4c, we have taken this effect into account by averaging the local density of states over the four edge positions. Clearly, the averaged edge protrusions are out of registry with respect to the basal plane S atoms, in agreement with the experimental observations.

Since we have come close to industrial conditions in our HDS experiment, we expect conversion of CH$_3$SH to CH$_4$[2,24]. Hence, the reaction should be in a steady state rather than in the static equilibrium discussed in the previous paragraph. To model how this affects the prevalent edge structure, we computed a reaction network linking a set of reaction intermediates on Au-supported MoS$_2$ (see Fig. 5). The catalytic cycles in the network consist of three stages: the conversion of CH$_3$SH to CH$_4$, leaving behind a sulfur atom on the MoS$_2$ edge, the desorption of this sulfur atom as H$_2$S, and the adsorption of hydrogen. From Fig. 5, it is clear that the conversion of CH$_3$SH is essentially a one-way reaction due to its high energy gain. In contrast, the adsorption/desorption steps of H$_2$, H$_2$S, and CH$_3$SH are all reversible.

Putting this in a simple steady state rate equation model we obtain:

$$CH_3SH_{ads} \rightarrow CH_{4(g)} + S_{ads} \tag{1}$$

$$S_{ads} + H_{2(g)} + CH_3SH_{(g)} \rightleftharpoons H_2S_{(g)} + CH_3SH_{ads} \tag{2}$$

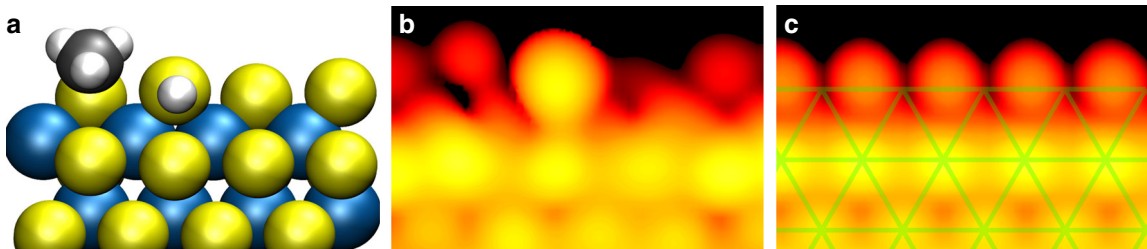

**Fig. 4** Thermodynamically preferred structure under the experimental desulfurization conditions (P$_{CH3SH}$ = 0.1 bar, P$_{H2S}$ = ~0.001 bar, P$_{H2}$ = 0.9 bar, 250 °C). **a** Ball model. **b** Simulated STM image for U$_s$ = −0.3 V, using an electron density contour value of 1 × 10$^{-6}$ AU. **c** Simulated STM image taking into account thermal averaging due to diffusion, adsorption and desorption. The green grid highlights the registry shift of the edge atoms with respect to the basal plane S atoms, which was also observed in the experiment

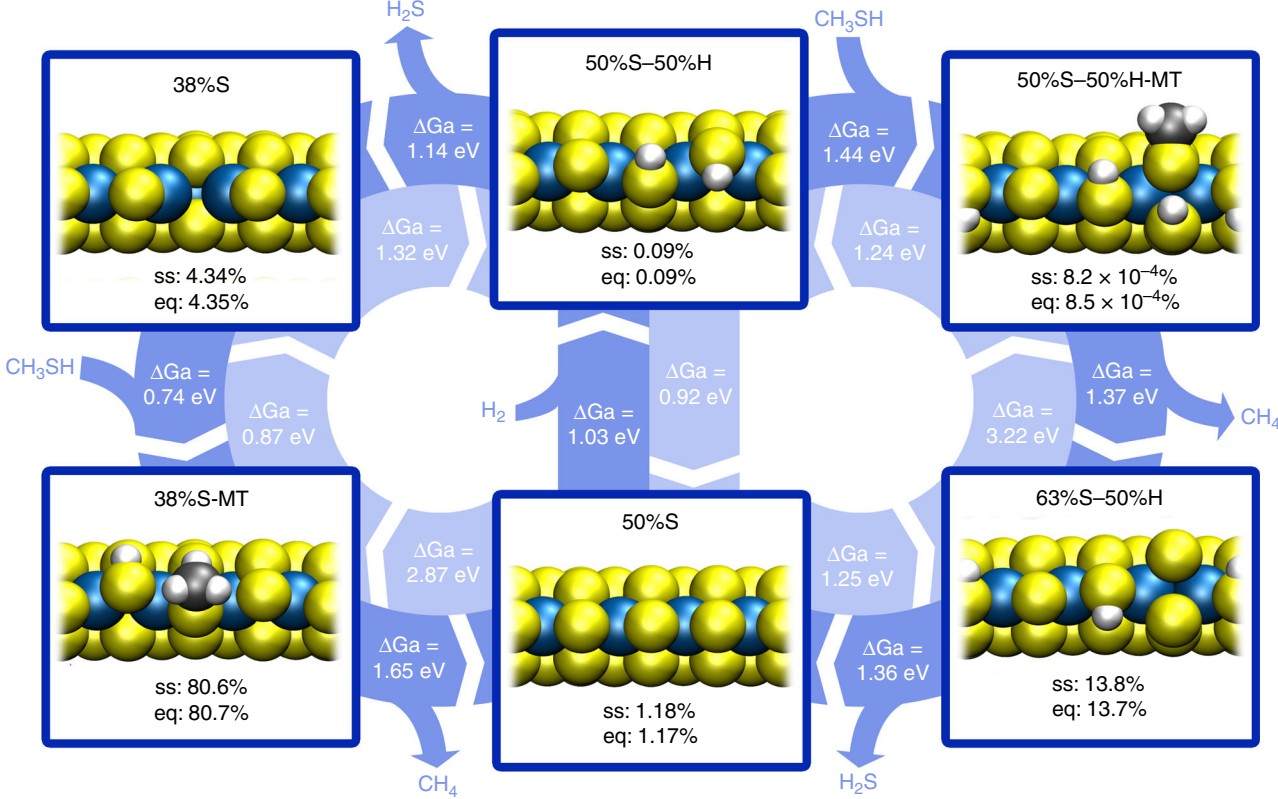

**Fig. 5** Reaction network for $CH_3SH$ desulfurization on $MoS_2$/Au(111). The activation free energy barriers ($\Delta G_a$) indicated in the arrows were calculated based on the experimental conditions ($P_{CH3SH} = 0.1$ bar, $P_{H2S} = {\sim}0.001$ bar, $P_{H2} = 0.9$ bar, 250 °C). The abundances of intermediates in steady state conditions and in an equilibrium where the C–S bond breaking step is disabled are designated by ss and eq, respectively. The steady state concentrations were derived using transition state theory and rate equation modeling (see Methods section). MT corresponds to methane thiol

$$\frac{[S_{ads}]}{[CH_3SH_{ads}]} = \frac{k_1 + k_{-2}P_{H_2S}}{k_2 P_{H_2} P_{CH_3SH}} \qquad (3)$$

Although highly simplified, this model provides some insight into the effect of $CH_3SH$ conversion on the $MoS_2$ edge structure. When the conversion faces a high barrier, rate constant $k_1$ will be low. In such a case, the edge structure should be close to the equilibrium of Eq. 2. In contrast, when the barrier for $CH_3SH$ conversion would be lower than the $H_2/H_2S$ adsorption/desorption barriers, one should expect that adsorbed $CH_3SH$ would be largely replaced by sulfur atoms.

Figure 5 shows that the C–S bond breaking barriers are slightly higher than the $H_2S$ desorption barriers. Because the rate constants have an exponential dependence on the energy barrier, this translates into orders of magnitude difference in rate. Indeed, when we quantify the abundances of all reaction intermediates (see "Methods" section), we still find an 80.6% abundance of the 38%S-$CH_3SH$ edge state, compared with 80.7% under equilibrium conditions. Hence, it appears that the structure observed in our HDS experiment is the 38%S-$CH_3SH$ edge state.

In a more general view, our theoretical analysis identifies two mechanisms that can steer the $MoS_2$ edge structure away from its equilibrium with $H_2$ and $H_2S$ during the HDS reaction. First, the adsorption of organic molecules may favor different edge S and H coverages. In our experiments, this leads to the counterintuitive observation that the edge S content is reduced due to $CH_3SH$ adsorption: the 63%S-50%H structure is favored in the absence of $CH_3SH$ adsorption, whereas the 38%S-25%$CH_3SH$ structure is the most stable one when we do take $CH_3SH$ adsorption into account.

This effect is likely also present for other industrially important reaction intermediates such as reduced thiophenes, which adsorb even stronger than $CH_3SH$[34]. Indeed, infrared spectroscopy has shown that thiophene adsorption is much more pronounced in a reducing atmosphere[35]. Other compounds were also found to adsorb under HDS conditions[36]. However, weakly adsorbing and/or sterically hindered (e.g., dimethyldibenzothiophene) organosulfur molecules may not have sufficient interaction with the $MoS_2$ edge to significantly alter the catalyst's resting state.

A second mechanism that steers the edge structure out of equilibrium is the deposition of sulfur via C–S bond scission. Although this appears to have only a minor effect in our experiment, subtle changes in the barrier for C–S bond breaking can have major consequences for the average edge structure. For instance, if the C–S barrier in the network in Fig. 4 were lowered by 0.3 eV, the 63%S-50%H structure would become dominant. Hence, support effects, the presence of defects such as corner sites, and the nature of the hydrocarbons that are desulfurized can all cause large variations in the average structure of $MoS_2$ under reaction conditions.

## Discussion

In summary, we have studied the catalytically active edge structure of $MoS_2$ nanoparticles on Au(111) in mixtures of $H_2$, $H_2S$, and $CH_3SH$ at temperatures up to 250 °C using a dedicated high-pressure scanning tunneling microscope. In hydrogen, trace amounts of sulfur in the feed are sufficient to maintain a sulfur coverage of 1 edge S atom per edge Mo atom. Surprisingly, the edge is reduced during the hydrodesulfurization of $CH_3SH$ to accommodate $CH_3SH$ adsorption. Due to the slow C–S bond scission on our model catalyst, the system remains close to an equilibrium state. However, our theoretical analysis indicates that small changes in the reaction rate or the

reaction mechanism, which could originate from support effects or from the presence of different hydrocarbons, can have a major influence on the average MoS$_2$ edge structure. In particular, for highly active MoS$_2$ catalysts one may expect that sulfur deposition by the conversion of organosulfur compounds increases the edge S coverage under hydrodesulfurization conditions, rather than to decrease it, as was found here. Hence, we conclude that the prevalent structure of the active sites during hydrodesulfurization catalysis on MoS$_2$ likely depends both on the precise type of catalyst and the nature of the feedstock.

## Methods

**Model catalyst preparation**. Clean, atomically smooth Au(111) was prepared by cycles of 1 keV Ar$^+$ bombardment and annealing at 627 °C. MoS$_2$ particles were deposited by evaporation of Mo in $1 \times 10^{-6}$ mbar H$_2$S, with the Au substrate at 150 °C, followed by annealing at 450 °C in $2 \times 10^{-6}$ mbar H$_2$S.

**High-pressure experiments**. The employed gases (Westfalen AG), Ar N5.0, H$_2$ N5.0, and CH$_3$SH N2.8 (main impurities dimethylsulfide and dimethyldisulfide) were fed through particle filters before use. Their purity was confirmed using mass spectroscopy. To prevent corrosion, the gas lines were made of Hastelloy C alloy, whereas the reactor consists of PEEK, Zerodur glass, and Kalrez. Measurements on the bare Au(111) surface under HDS conditions confirmed the absence of impurity deposition other than the slow formation of a sulfur overlayer. All gas lines were flushed with argon for at least 30 min prior to each experiment. To start the high-pressure exposure, the reactor was slowly pressurized in H$_2$ and subsequently heated to the desired temperature. For the HDS experiments, CH$_3$SH was mixed in the reactor feed only after reaching 250 °C. In order to minimize the thermal drift in the microscope, the system was allowed an equilibration period of ~90 min. To cope with the remaining drift, image acquisition times of around 20 s per image were employed and a drift correction was applied (making use of the well-defined 60/120 degree angles in the particles). As the noise level increased under high-pressure, high-temperature conditions, a $3 \times 3$ pixel averaging was applied for clarity.

**Theoretical analysis**. All DFT calculations were carried out with the BAND program package[37–41], using the PBE density functional[42], Grimme van der Waals corrections[43], and scalar relativistic corrections. A triple-ζ plus polarization basis set was used for the valence orbitals, while the core orbitals were kept frozen in the same state as in the free atoms. In general, default settings of the BAND program were used. The Self-Consistent Field convergence criterion was set at $10^{-6}$ Hartree atomic units, while the geometrical optimization criterion was set at $10^{-2}$ Hartree per nanometer.

A ($4 \times 4$) MoS$_2$ unit cell was employed, in the form of a stripe with periodicity in one direction. The stripe contains both an S-type edge and an Mo-type edge. The S-type edge was kept fully covered in all calculations (2 S edge atoms per Mo edge atom). The length of the unit cell was kept at the value optimized for 50%S coverage: 1.248 nm. For the calculations where the gold support was included, the Au(111) surface was modeled by a 2 layer slab with a lattice parameter commensurate with the MoS$_2$ stripe. The S-Mo-S-Au-Au stacking was chosen as A-B-A-B-C, with a S-Au layer spacing of 0.442 nm. The number of k points chosen for sampling the Brillouin zone was 3 throughout.

Transition states were located as follows: for a chosen reaction coordinate (usually an interatomic distance) total energies were computed for a range of fixed values (while optimizing all other degrees of freedom). For the structure of highest energy a partial hessian was calculated, including atoms at or close to the reaction site. The most negative eigenvalue of this partial hessian was used to locate the saddle point. In most cases the search had to be restarted several times by recomputing the partial hessian on the structure with the smallest gradient found so far. It was checked that the partial hessian of the final structure had precisely one negative eigenvalue.

The reaction energies ($\Delta E_r$) used in the computation of phase diagrams were calculated per unit cell as:

$$\Delta E_r = E_{MoS_2,S_xH_y} - E_{MoS_2} - x\,E_{H_2S} - \left(\tfrac{1}{2}y - x\right)E_{H_2} \tag{4}$$

In Eq. 4, E$_x$ denotes the total electronic energy obtained from DFT for the respective structure. In order to calculate the free energy of an edge structure, entropic corrections need to be applied:

$$\Delta\mu_S = RT \ln\left(\frac{P_{H_2S}}{P_{H_2}}\right) - T\left(S^o_{H_2S} - S^o_{H_2}\right) \tag{5}$$

$$\Delta\mu_H = \frac{1}{2}\left(RT \ln\left(P_{H_2}\right) - T\,S^o_{H_2}\right) \tag{6}$$

The standard entropies (S$^o_x$) in these equations were obtained from thermodynamic tables[44]. Using the entropic corrections, one can compute the free

energy change involved in a reaction as:

$$\Delta G_r = \Delta E_r - x\,\Delta\mu_S - y\,\Delta\mu_H \tag{7}$$

The structure with the lowest free energy per unit cell at a particular combination of $\Delta\mu_S$ and $\Delta\mu_H$ values will be the phase present under those conditions. Note that with this definition of the free energy, we assume that all solid phases have zero entropy and that there is no change in heat capacity during the reaction.

To model the adsorption of CH$_3$SH, we first calculated the adsorption energy:

$$\Delta E_{ads} = E_{MoS_2,S_xH_y,MT_z} - E_{MoS_2,S_xH_y} - z\,E_{MT} \tag{8}$$

In Eq. 8, MT corresponds to CH$_3$SH. The thermodynamic stability of the adsorbed CH$_3$SH structure was calculated as its formation free energy with respect to the most stable phase in the absence of CH$_3$SH adsorption:

$$\Delta G_f = \Delta E_{ads} - z\,\Delta\mu_{MT} + \Delta G_{r-MoS_2,S_xH_y} - \Delta G_{r-MoS_2,S_nH_m} \tag{9}$$

$$\Delta\mu_{MT} = RT \ln(P_{MT}) - TS^o_{MT} \tag{10}$$

In Eq. 9, MoS$_2$,S$_n$H$_m$ is the most stable structure in the absence of CH$_3$SH adsorption.

Finally, we performed a kinetic analysis using the standard free energy barriers that link the reaction intermediates. Entropic corrections (for standard conditions) were again applied using Eqs. 5, 6, and 10. Rate constants were calculated from the standard free energy barriers ($\Delta G^o_a$) using transition state theory:

$$k = \frac{k_B T}{h} e^{\frac{-\Delta G^o_a}{k_B T}} \tag{11}$$

Here, we assume that there is no communication between adjacent unit cells. Furthermore, it is assumed that only the six unit cell structures in Fig. 5 can be formed. When the reaction reaches steady state, the coverages of all six structures will be fixed. Thus, we obtain a set of linear equations, shown here for the example of an intermediate with two links to other intermediates.

$$\frac{d\theta_n}{dt} = k_{n-1}\theta_{n-1} + k_{-n}\theta_{n+1} - \left(k_{-(n-1)} + k_n\right)\theta_n = 0 \tag{12}$$

In Eq. 12, $n$ designates a particular structure, whereas $n-1$ and $n+1$ are the structures before and after structure $n$ in the reaction chain, respectively. Reactions in the backwards direction are indicated by a minus sign (e.g., $k_{-n}$). By solving this set of equations, one obtains the steady state coverage of the various intermediates. For a more detailed description of the model, see Supplementary Note 5.

## Data availability

The data that support the findings in this study are available from the corresponding author upon reasonable request. The STM images shown in the paper, the total energies of the DFT of all considered edge structures (with and without Au support), and gas phase molecules and the local density of state (summed between 0 eV and −0.3 eV vs. the Fermi level) are provided as a Source Data file. Only commercially available software was used for the analysis and representation of data.

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

## Acknowledgements
This project was financially supported by a Dutch SmartMix grant and by NIMIC partner organizations through NIMIC, a public–private partnership. I.M.N.G. acknowledges the Dutch organization for scientific research for her Veni fellowship. The Dutch organization for scientific research is also thanked for providing computing time on the Cartesius facility (grant numbers 15283 and SH-325-15). The authors thank Dr. Stig Helveg (Haldor Topsøe company), Dr. Bart Nelissen (Albermarle Corporation) and Prof. Dr. Eelco Vogt (Albemarle Corporation/Utrecht University) for fruitful discussions.

## Author contributions
R.V.M., I.M.N.G. and J.W.M.F. designed the STM experiments. J.N.L. preformed the DFT calculations. R.V.M. performed the STM experiments, the thermodynamic analysis and wrote the paper. All authors contributed to discussions and interpretation of the data.

## Additional information

**Competing interests:** The authors declare no competing interests.

