## [Peer Review File · Nature Communications]

Reviewers' comments:

Reviewer #1 (Remarks to the Author):

The paper by Mom et al is a combined experimental and theoretical study of MoS₂ nanocrystals on Au(111) which is a model system for the Mo-based hydrodesulfurization catalysts used for reducing S levels in fossil fuels. The authors use their proven high-pressure reactor STM system to image the model system under conditions that are close to industrial conditions, in the intention to determine the structure and sulfur coverage on the edges of MoS₂ nanocrystals under real working conditions. The authors claim from their STM images that the sulfur coverage drops from the as-synthesized situation due to the H₂ gas environment, where introduction of the simplest possible S-containing hydrocarbon (CH₃-SH) apparently leads to a more complex situation. Theory is employed to explain the observations further, and model calculations that take the substrate into account shows expectedly that the transition from full S coverage to a lower sulfur coverage takes place upon H introduction. This is confirming numerous theoretical predictions and "averaging" in-situ TPD (J. Catal. 280, 178 (2011)) and spectroscopy observations (well cited in the paper). The same edge structures with reduced S coverage have also been imaged in regular UHV STM studies previously resulting from H₂ exposure (e.g J. Catal. 221, 510 (2004)). Next, the authors discuss how CH₃-SH adsorbs on the edge and a kinetic analysis suggests that CH₃-S species apparently builds up at a high concentration on the edge due to a high barrier for C-S bond breaking. The overall conclusion is that relevant active edge structures can be accessed using the reactor STM approach, but the authors also discuss that more complex situations need to be considered to eventually compare with industrial catalytic conditions.

The application of the reactor STM for this kind of work in hydrodesulfurization is a very strong achievement on a very high technical level, and is a valuable next step in model studies of this important catalyst. Unfortunately the STM data set produced is not very extensive, which is probably due to the significant challenges involved, and this also influences the overall strength of the conclusions (see below). It may be of general interest to the readers of a multidisciplinary journal such as Nature Communications to observe that in-situ scanning probe microscopy has improved to the level where sulfidic systems can be studied. This is a spectacular achievement, but in terms of new findings in the particular field of hydrodesulfurization catalysis, the paper is less significant at this point. Furthermore the analysis related to comparison between STM experiment and modelling has some problems that need to be addressed.

- One of the difficulties in the comparison between experiment and STM simulations is that basically all fine structure predicted in simulations is argued to be smeared out due to fast diffusion of species along the edge at the imaging temperature. While these assignments are quite reasonable given the temperature, the overall determination of the exact edge structure using this imaging method becomes somewhat speculative since such a scenario will fit with any suggested edge structure. Here a more extensive set of experiments could have been helpful in supporting the conclusions, i.e. temperature series, or imaging quenched samples in vacuum afterwards.
- The comparison between STM images to determine the changes in position of edge protrusions between Figure 2a-c needs to be quantitative in order to become convincing. The image quality makes it hard to judge from the illustrations, so further quantitative measures such as line scans and distances need to be included and compared with the simulations. It is hard to discern exactly what the authors mean by edge protrusions. The authors use the terms "in" and out of "registry", but a 2D grid placed on the particle is needed to determine this in detail instead of just a single guide line. Since so much of the paper scope relies on the observation that something happens between figure 2a, b and c when the authors claim to see an effect of the CH₃SH, this important aspect needs to be stronger presented.
- STM image quality needs to be discussed in general, in particular in relation to the resolution on the edges, as the images presented here appear show different types of contrast within the same particles (Figure 1, and large images in figure 2). Only small averaged sections are discussed, but how robust are the conclusions when applied to the other edge of the same particles within a

dataset?

- Theory and also recent IR and TEM results (ACS Catal. 6, 1081 (2016)) indicate that the MoS₂ shape itself should be a function of the conditions, i.e. that the somewhat regular triangular shape in Fig 1 should change to a more hexagon like shape when exposed to hydrogen. Was there any indication in the experiment to address this effect?

- Previous STM images (e.g. Phys. Rev. Lett. 87, 196803 (2001)) have resolved bright stripes adjacent to the edge (claimed to be due to metallic states), but these seem much differently presented here or even absent. Is this an effect of the STM imaging conditions or due to H₂ at elevated pressure?

- The calculations use a Au surface which is stretched to match the MoS₂ lattice. Is the Au underneath the MoS₂ also stretched in the experiment? This is a significant elongation of the bonds and correspondingly this would make the Au surface in the model more reactive. What influence does this have on the conclusions and the equilibrium coverages?

- It is mentioned that a (1x1) Au(111) surface is imaged during exposure (figure 1b). Does this mean that the herringbone reconstruction on the Au(111) surface is lifted due to the thiols at these pressures? In addition, I am puzzled about the long term build-up of S on the Au? How does this structure compare with previously proposed S-adlayer structures on Au? And was an experiment without MoS₂ on the surface done to correlate if it is entirely due to the CH₃SH?

- I am missing information on what was done to ensure purity of the gases supplied? Since H₂S is corrosive, there is a need to select materials in the gas systems to prevent transport of impurities. Also to which extent does the CH₃SH decompose before it reaches the sample? Was some kind of spectroscopy (e.g. XPS) used to confirm sample purity after the experiment?

- It is highlighted in the paper that a fractional coverage of 63% S and 50% H is surprisingly found to be stable under imaging conditions, which contrasts previous findings. Most previous studies looked at less wide unit cells, and the emergence of the claimed new structures here is most likely due to the better resolution of a 4 unit cell wide model. The trends are in reality the same, and I suppose if one used a 5 unit cell wide model, one would see and even more gradual transition between high S and low S coverages and a more complicated phase diagram. In addition, it is likely that corners of a particle may play a role here (as also discussed in the paper), which could change the picture altogether.

- How does the claimed presence of CH₃-S species adsorbed on the edges compare with literature? There are very extensive operando IR studies on the catalysts system, so it would be interesting if additional support exists in the literature for such stable hydrocarbon species. Some studies also report on Mo-carbide formation under working conditions, but is the structure proposed here compatible with this idea?

Reviewer #2 (Remarks to the Author):

Using advanced STM techniques combined with DFT calculations, the authors have performed a detailed investigation toward detection of the in-situ variation of active edge site of MoS₂ under CH₃SH hydrodesulfurization conditions. After careful evaluation of the manuscript, I think the developed STM techniques are state-of-the-art, the computational methods are reasonable. Moreover, the paper is novel and the conclusions are original which will be of interest to others in the broad community. Thus, I suggest the paper should be published in Nature communications eventually.

However, I have three comments and the authors should be cautiously considered at this stage. The first one is, the phase diagram of MoS₂ edge structures in H₂/H₂S mixtures have been extensively explored in previous studies. Although the authors listed the result offsets with those of published results in the supporting information, a detailed comparison should be given. For examples, what is the reference of sulfur structures in these literatures, how do they get these offsets and the relative energies, and what is the difference with the phase diagrams reported previously, i.e., the emergence of 63% and 38% S.

Secondly, the authors use CH₃SH as a model compound to simulate the S-contained species in oil.

However, the compositions of oil are rather complicated, depending on many factors, such as place of origin, etc. The authors should comment whether CH₃SH has a representativeness for S-contained oil. I am wondering what the situations are for other typical model compounds, such as thiophene, tetrahydrothiophene and other refractory S-contained species. Similar question is, the authors claimed that they use a dedicated high-pressure STM, but the pressure is still largely lower than the realistic condition. I don't know whether it can represent the real situation. Finally, the rates calculated by formula (12) seem questionable. If two gases involved in the elementary reaction, two coverages of theta should be used. K_n and K_{n-1} should be explained in the text. I suggest the kinetic parts should be fully described.

Reviewer 1

- One of the difficulties in the comparison between experiment and STM simulations is that basically all fine structure predicted in simulations is argued to be smeared out due to fast diffusion of species along the edge at the imaging temperature. While these assignments are quite reasonable given the temperature, the overall determination of the exact edge structure using this imaging method becomes somewhat speculative since such a scenario will fit with any suggested edge structure. Here a more extensive set of experiments could have been helpful in supporting the conclusions, i.e. temperature series, or imaging quenched samples in vacuum afterwards.

We thank the reviewer for this comment, which requires some additional evidence. This can, however, not be obtained through a temperature series or quenching for two reasons: first, changing the temperature will result in different thermodynamic and kinetic conditions, which may well change the edge state that is obtained. To give an extreme example: the 100%S Mo edge cannot be reduced using pure H₂ in ultrahigh vacuum due to kinetic limitations. Higher temperatures in our ReactorSTM setup are not an option, as we are already close to the limit of the chemical and thermal stability of our Kalrez reactor seal. The second reason is that CH₃SH adsorption and (rapid) decomposition on Au(111) occurs at temperatures (slightly) lower than probed here. This will result in the formation of a sulfur overlayer on the Au(111) substrate. The rate at which this occurs is much higher at intermediate temperatures than at our reaction temperature due to the increased CH₃SH coverage on the surface. The edge sites of MoS₂ particles surrounded by a sulfur overlayer are exceedingly difficult to probe, and may be altered due to the presence of the overlayer.

That said, we would like to strengthen our argumentation by explicit simulation of the temperature-averaged STM images. In the main text, we have added a temperature-averaged simulation of the 38%S-25% CH₃SH structure to Figure 4. This simulation clearly shows that the edge protrusions do not follow the registry of the basal plane atoms, similar to our experimental observations. We also performed temperature averaging for the 50%S-50%H structure in Figure S2. Again, this simulation corroborates our conclusion that the 50%S and 50%S-50%H structures look very similar to each other, yet distinctly different from the 38%S-25% CH₃SH structure.

- The comparison between STM images to determine the changes in position of edge protrusions between Figure 2a-c needs to be quantitative in order to become convincing. The image quality makes it hard to judge from the illustrations, so further quantitative measures such as line scans and distances need to be included and compared with the simulations. It is hard to discern exactly what the authors mean by edge protrusions. The authors use the terms “in” and out of “registry”, but a 2D grid placed on the particle is needed to determine this in detail instead of just a single guide line. Since so much of the paper scope relies on the observation that something happens between figure 2a, b and c when the authors claim to see an effect of the CH₃SH, this important aspect needs to be stronger presented.

Following the opportunities for stronger presentation suggested by the reviewer, we have implemented a 2D grid in Figures 2, 4, and S1-4. To clarify that terms “in registry” and “out of registry” refer to the position of the edge protrusions with respect to the basal plane atomic registry we added the following sentence in the last paragraph of page 3: *“Indeed, Figure 2a shows that the*

apparent location of the edge atoms does not follow the registry of the basal plane atoms.” To ensure clarity of the term “edge protrusions” we added the sentence: “To clearly separate the nomenclature for apparent and actual edge atom positions, we refer to the apparent positions of the edge atoms as edge protrusions hereafter.”

We should point out that further quantitative analysis of the data will likely lead to overinterpretation, as the noise level in the tunneling current due to fast adsorption/desorption processes on the tip apex induce prohibitively high noise levels in the tunneling current. However, as also discussed in the next point, we have added additional analyses of edge structures in the supporting information to provide a more confident assignment.

- STM image quality needs to be discussed in general, in particular in relation to the resolution on the edges, as the images presented here appear show different types of contrast within the same particles (Figure 1, and large images in figure 2). Only small averaged sections are discussed, but how robust are the conclusions when applied to the other edge of the same particles within a dataset?

We thank the reviewer for raising this point, which we should further clarify. In regular STM experiments, the tip is often conditioned until the tip apex is symmetrical. Once a symmetrical apex is obtained, it will usually remain in this good state for a prolonged amount of time. Under our high-pressure, high-temperature conditions however, frequent changes in the tip apex are unavoidable. Therefore, obtaining an extremely sharp *and* symmetric apex for a period long enough to acquire an image is virtually impossible, even with a PtIr tip. Hence, the images show some asymmetry, often showing sufficient resolution on only part of the edges. In the manuscript, this point is addressed at the top of page 4: “Note that some tip asymmetry could not be avoided under high-pressure, high-temperature conditions, resulting in somewhat asymmetrical image sharpness. To prevent misinterpretation, the conclusions from Figure 2b were corroborated using additional data (see Figure S3).” To ensure correct interpretation of the edge structure under HDS conditions, we followed the reviewer’s suggestion and analyzed the alternative edge of Figure 2c. The result is shown in Figure S4 and discussed in the neighboring paragraph.

- Theory and also recent IR and TEM results (ACS Catal. 6, 1081 (2016)) indicate that the MoS₂ shape itself should be a function of the conditions, i.e. that the somewhat regular triangular shape in Fig 1 should change to a more hexagon like shape when exposed to hydrogen. Was there any indication in the experiment to address this effect?

Upon the start of our experiments, we had the same expectation as the reviewer. To test for shape changes, we exposed as-prepared particles to hydrogen for several hours. We counted the fraction of pure triangular particles before and after this exposure, but did not find a convincing indication for any shape change. We attribute this to kinetic limitations, that were also observed in somewhat similar experiments by Lauritsen *et al.* Journal of Catalysis 221 (2004) 510–522. In the meantime, the same group has shown that shape change can be induced at higher temperatures. As this point is clearly of interest, we have added it to the first paragraph of page 4.

- Previous STM images (e.g. Phys. Rev. Lett. 87, 196803 (2001)) have resolved bright stripes adjacent to the edge (claimed to be due to metallic states), but these seem much differently presented here or even absent. Is this an effect of the STM imaging conditions or due to H₂ at elevated pressure?

The bright stripes at the particle edges were usually observed, independent of the gas environment, although their intensity appears to depend on the tip structure (adsorbates on the apex) and likely also on the edge state. However, we have shown mostly differentiated images in the manuscript, in which the bright stripes appear much weaker. We have now stressed this in the last paragraph of page 3: “We should also point out that the STM images in Figure 2 were differentiated to highlight the atomic contrast. In this display method, the metallic “bright” edge states that are usually observed at the edges of the MoS₂ particles are less apparent (see also Figure S5 in the Supporting Information).” Furthermore, we note in the second paragraph of page 4 that: “In agreement with the observations on reduced MoS₂ edge structures in the literature, we note that the “bright” metallic edge states are maintained (see Figure S5 in the Supporting Information)”. Figure S5 shows the image in Figure 2b as a regular line-by-line background subtracted image rather than in differentiated display.

- The calculations use a Au surface which is stretched to match the MoS₂ lattice. Is the Au underneath the MoS₂ also stretched in the experiment? This is a significant elongation of the bonds and correspondingly this would make the Au surface in the model more reactive. What influence does this have on the conclusions and the equilibrium coverages?

The reviewer raises a just point here. We noted that we had not referred to Section 2 in the SI in the main text, which discusses the effect of the Au(111) support. We have added this reference in the discussion of Figure 3 and addressed the overestimation of the support effect due to the stretching of the Au(111) substrate in section S2 of the Supporting Information. We have also extended this discussion to the adsorption of CH₃SH in Supporting Information Section S4.

- It is mentioned that a (1x1) Au(111) surface is imaged during exposure (figure 1b). Does this mean that the herringbone reconstruction on the Au(111) surface is lifted due to the thiols at these pressures? In addition, I am puzzled about the long term build-up of S on the Au? How does this structure compare with previously proposed S-adlayer structures on Au? And was an experiment without MoS₂ on the surface done to correlate if it is entirely due to the CH₃SH?

We have intensively studied the interaction of CH₃SH and Au(111) (more publications will follow), and added more details based on the importance that the reviewer points out here. Page 3, 3rd paragraph: *“However, the absence of the herringbone reconstruction observed on clean Au(111) [Barth et al., PRB 42, 1990] indicates that some (dissociated) CH₃S is present on the surface. Decomposition of CH₃S or H₂S leads to the formation of a sulfur layer over time (see Figure 1c), with a structure resembling that observed by Lay et al. [Lay et al., Langmuir 19, 2003]. The formation of the sulfur overlayer occurred independent on whether MoS₂ particles were present on the Au(111) substrate or not.”*

- I am missing information on what was done to ensure purity of the gases supplied? Since H₂S is corrosive, there is a need to select materials in the gas systems to prevent transport of impurities. Also to which extent does the CH₃SH decompose before it reaches the sample? Was some kind of spectroscopy (e.g. XPS) used to confirm sample purity after the experiment?

The reviewer raises an important point here, which we addressed experimentally, yet did not discuss in the manuscript. We have added the following in the methods section: *“The employed gases (Westfalen AG), Ar N5.0, H₂ N5.0 and CH₃SH N2.8 (main impurities dimethylsulfide and dimethyldisulfide) were fed through particle filters before use. Their purity was confirmed using mass spectroscopy. To prevent corrosion, the gas lines were made of Hastelloy C alloy, whereas the reactor consists of PEEK, Zerodur glass and Kalrez. Measurements on the bare Au(111) surface under HDS conditions confirmed the absence of impurity deposition other than the slow formation of a sulfur overlayer”.*

- It is highlighted in the paper that a fractional coverage of 63% S and 50% H is surprisingly found to be stable under imaging conditions, which contrasts previous findings. Most previous studies looked at less wide unit cells, and the emergence of the claimed new structures here is most likely due to the better resolution of a 4 unit cell wide model. The trends are in reality the same, and I suppose if one used a 5 unit cell wide model, one would see and even more gradual transition between high S and low S coverages and a more complicated phase diagram. In addition, it is likely that corners of a particle may play a role here (as also discussed in the paper), which could change the picture altogether.

We agree in part with the reviewer on this point. On the one hand, a larger unit cell may indeed yield an even larger variety of stable edge structures (as stressed in the first paragraph on page 5: *“However, we should point out that for a larger unit cell size, an even larger variety of structures may appear in the phase diagram.”*). On the other hand, the transition from 0%S to 100%S is not gradual over the whole range of chemical potentials. As pointed out in the second paragraph of page 5, many phases are “skipped” in the phase diagram, because the intermediate coverage is never more stable than either the lower or the higher coverage. For instance, 50%S, 50%H is missing in the phase diagram for Au-supported MoS₂. The effect is even stronger at high S coverage, where the 75% S and 87% S are not present. Hence, we do feel that the stability of the 63%S and 50%H phase, in particular with respect to the 50%S phases, is surprising and meaningful. To make it more clear for the reader why this phase has not been found in the literature, we extended our literature analysis in section S3 in the ESI: *“We should note that there were also differences in the employed unit cells and the range of investigated structures. Where Bollinger et al. and Lauritsen et al. used a 2-Mo-atom-wide unit cell, Cristol et al. employed a 3-Mo-atom-wide unit cell and Prodhomme et al. a 4-Mo-atom-wide cell. Obviously, larger unit cells lead to more flexibility in the observed structures. The low-symmetry 38%S-x%H and 63%S-x%H structures which we have found to be stable over a wide range of conditions were not accessible to Bollinger et al., Lauritsen et al. and Cristol et al.. Prodhomme et al. did use the same unit cell size as in the present work, but did not consider structures with a coverage higher than 50%S. On the other hand, they did consider the 38%S structures and found their stability somewhat lower than in our work: the transition from 38%S to 50%S occurs already at $\Delta\mu_S = -0.76$ eV, whereas our calculations (without Au support) put the transition at -0.51 eV. This difference could be*

related to the differences in used methodology (VASP/projector augmented waves for Prodhomme et al. versus the orbital-based basis set used in the BAND package employed here). The orbital-based basis set used in our case is particularly suited for the simulation of situations where the electron density shows strong gradients, as is the case with the irregular 38%S-x%H and 63%S-x%H structures. This could lead to a higher stabilization of these structures.”

- How does the claimed presence of CH₃-S species adsorbed on the edges compare with literature? There are very extensive operando IR studies on the catalysts system, so it would be interesting if additional support exists in the literature for such stable hydrocarbon species. Some studies also report on Mo-carbide formation under working conditions, but is the structure proposed here compatible with this idea?

We thank the reviewer for this suggestion, which allows us to further substantiate our claims. While no direct comparison to DFT or IR studies for CH₃SH on Au-supported MoS₂ is available, some relevant studies on unsupported catalysts have been conducted. We have added the CH₃SH adsorption energies for unsupported MoS₂ in section S4 in the ESI and show good agreement with literature DFT. Furthermore, we have provided references to in situ and ex situ IR spectroscopy showing that adsorption of hydrocarbons in H₂-hydrocarbon mixtures is a general phenomenon that can be expected to occur over a wide range of HDS conditions, and that adsorption is promoted by having a reducing atmosphere (see the first paragraph on page 8).

We agree with the reviewer that the possibility of Mo-carbide formation on industrial HDS catalysts should not be excluded, given the clear evidence that MoS₂ is prone to form Mo₂C under certain conditions (Zhang et al., *Int. J. of Min. Met and Mat.* 25, 2018, Kelty et al., *App. Cat. A* 322, 2007, Jeon et al., *ACS Nano* 12, 2018). However, it seems that carbon-rich, sulfur-poor conditions are necessary and the reaction is usually carried out at high temperatures. Hence, Mo₂C formation under our sulfur-rich, low-temperature conditions seems unlikely. This point is now made in the first paragraph on page 5: “Mo carbide formation can be excluded, since our low-temperature, sulfur-rich HDS environment is far away from the conditions for which Mo₂C or MoS_xC_y formation were observed³⁰⁻³².”

Reviewer 2

The first one is, the phase diagram of MoS₂ edge structures in H₂/H₂S mixtures have been extensively explored in previous studies. Although the authors listed the result offsets with those of published results in the supporting information, a detailed comparison should be given. For examples, what is the reference of sulfur structures in these literatures, how do they get these offsets and the relative energies, and what is the difference with the phase diagrams reported previously, i.e., the emergence of 63% and 38% S.

To improve the completeness of the literature comparison, we have listed the reference and unit cell size used in the cited papers in section 3 of the supporting information. A detailed discussion on the emergence of the 38% and 63% structures is also provided.

Secondly, the authors use CH₃SH as a model compound to simulate the S-contained species in oil. However, the compositions of oil are rather complicated, depending on many factors, such as place of origin, etc. The authors should comment whether CH₃SH has a representativeness for S-contained oil. I am wondering what the situations are for other typical model compounds, such as thiophene, tetrahydrothiophene and other refractory S-contained species. Similar question is, the authors claimed that they use a dedicated high-pressure STM, but the pressure is still largely lower than the realistic condition. I don't know whether it can represent the real situation.

We thank the reviewer for raising these questions, as it appears we have not sufficiently stressed the important discussion of representativeness. We have addressed this in the second paragraph on page 3: *"Naturally, a single organosulfur compound can never fully represent the complex mixture in oil. However, mercaptans such as CH₃SH constitute a major component in crude oil²³. Our applied temperature and pressure (250 °C, 1 bar) are close to the typical hydrodesulfurization conditions applied for the light naphtha fraction (260-380 °C, 5-10 bar)² and are sufficient to achieve catalytic turnover²⁴."* Additionally we discuss in the first paragraph on page 8: *"In our experiments, this leads to the counterintuitive observation that the edge S content is reduced due to CH₃SH adsorption: the 63%S-50%H structure is favored in the absence of CH₃SH adsorption, whereas the 38%S-25%CH₃SH structure is the most stable one when we do take CH₃SH adsorption into account. This effect is likely also present for other industrially important reaction intermediates such as reduced thiophenes, which adsorb even stronger than CH₃SH³³. Indeed, infrared spectroscopy has shown that thiophene adsorption is much more pronounced in a reducing atmosphere³⁴. Other compounds were also found to adsorb under HDS conditions³⁵. However, weakly adsorbing and/or sterically hindered (e.g. dimethyldibenzothiophene) organosulfur molecules may not have sufficient interaction with the MoS₂ edge to significantly alter the catalyst's resting state."*

Finally, the rates calculated by formula (12) seem questionable. If two gases involved in the elementary reaction, two coverages of theta should be used. Kn and Kn-1 should be explained in the text. I suggest the kinetic parts should be fully described.

Based on the remarks of the reviewer, it is clear that we have insufficiently described the kinetic model. To aid the reader, an extensive description of the model has now been added to the supporting information (section 5).

Reviewers' comments:

Reviewer #1 (Remarks to the Author):

The authors have revised the manuscript in several ways, and it has been significantly improved. One of the problematic issues in the original version was the somewhat weak link between STM experiment and the structures proposed from theory. This part has been improved, and the authors have especially strengthened the argumentation of their structures in relation to kinetically stabilized structures providing time-averaged STM images. The paper is a good contribution demonstrating the use of in-situ atomic-scale STM imaging of complex catalyst materials such as a metal sulfide, and it should be published once the authors have clarified the following:

While it is significant strength that the authors have more critically assessed the STM comparison with simulations, some parts of the image correspondence with simulations may still be ambiguous. The authors ought to describe more details on how they observe the correspondence directly in the paper.

1. The assignment of protrusions being 'in' and 'out' of registry should be better defined. The positions marked in the overlaid lattice are placed asymmetrically with respect to the lattice and not very far from lattice positions, which is not something immediately seen in the simulations? It would be appropriate to address possible reasons for the difference directly in the paper. Essentially, the observant reader may be left with the impression that the STM resolution is, in fact, not sufficient to conclude the way the authors do?

2. Comparison of figure 4 with the STM image in figure 2 suggest that the edge protrusions are in fact in front of the bright stripe, and therefore not in good correspondence with the assigned lattice. Can this be clarified?

3. Also, regarding the slight shift of the position of edge protrusions relative to the lattice: It is not clear if this is analyzed directly on differentiated images displayed in Fig 2. If so, this could potentially be a problem. For an ideal lattice differentiation is unproblematic, but for the more complicated contrast on the edges in the STM images and depending on the direction of the gradient used, I am in doubt if this could affect the relative in plane position, especially as edge protrusion are less corrugated here. The authors need to rule out that the very small shift, which is crucial to their interpretation in Fig 2a and c, is not related with the image-processing procedure.

Reviewer #2 (Remarks to the Author):

After a considerable long-time modification, the quality of the manuscript was greatly improved. In particular, all my raised concerns were positively responded, therefore, I'd like to recommend its publication in Nat. Communications.

Yucheng Huang

Reviewer #1

1. The assignment of protrusions being ‘in’ and ‘out’ of registry should be better defined. The positions marked in the overlaid lattice are placed asymmetrically with respect to the lattice and not very far from lattice positions, which is not something immediately seen in the simulations? It would be appropriate to address possible reasons for the difference directly in the paper. Essentially, the observant reader may be left with the impression that the STM resolution is, in fact, not sufficient to conclude the way the authors do?

We thank the reviewer for identifying this insufficiently convincing treatment, which we now discuss more extensively and quantitatively.

On page 3, we note regarding ash prepared particles:

“The observed registry shift is smaller than the expected half lattice spacing, which was also observed in some cases in the literature^{9,15,28}. We attribute this to asymmetry in the tip apex, which can lead to slight distortions in the observed edge structure.”

Regarding the edge structure in hydrogen, we note on page 4:

“In all cases, we find that the edge protrusions are located precisely in registry (on average, we measure 2% of a lattice spacing shift measured with respect to the basal plane registry).”

This calculation was performed using the data in Figures 2b and S4.

Regarding the structure under HDS conditions, we added on page 5:

“Figure 2c shows that the edge structure under hydrodesulfurization conditions has changed with respect to the structure in pure hydrogen (see also Figure S5 and see Figure S6 for the alignment procedure of the grid overlay). The observed registry shift of the edge protrusions is similar to that observed for the 100%Si covered edge in Figure 2a: on average 20% of a lattice spacing out of registry in Figure 2a and 18% in Figures 2c and S5. Again, we note that the registry shift is asymmetric with respect to the lattice, which we attribute to asymmetry in the tip apex. Indeed, one can observe that in both Figure 2a and 2c the resolution on the basal plane atoms is higher for the horizontal direction than for the vertical direction, which explains why the two edge structures show the same asymmetry.”

2. Comparison of figure 4 with the STM image in figure 2 suggest that the edge protrusions are in fact in front of the bright stripe, and therefore not in good correspondence with the assigned lattice. Can this be clarified?

Based on the reviewer’s comment, it is clear that our procedure for determining the lattice registry was insufficiently described. In the derivative images, the position of the bright stripe is

not very clear. This can for example be seen in the newly added Figure S3 in the supporting information, which shows both the differentiated and non-differentiated version of the data in Figure 2a, and similarly for Figure 2b and Figure S7. Hence, comparing Figure 2c and Figure 4 in this aspect could be misleading. To confirm that the placement of the grid is correct, we have now added Figure S6 in the supporting information and note in its description:

“Note that the overlay grids in Figures S5 and 2c were placed based on the alignment of a larger grid matching the basal plane lattice, the particle’s corner, and both edges at the same time (see Figure S6).”

3. Also, regarding the slight shift of the position of edge protrusions relative to the lattice: It is not clear if this is analyzed directly on differentiated images displayed in Fig 2. If so, this could potentially be a problem. For an ideal lattice differentiation is unproblematic, but for the more complicated contrast on the edges in the STM images and depending on the direction of the gradient used, I am in doubt if this could affect the relative in plane position, especially as edge protrusion are less corrugated here. The authors need to rule out that the very small shift, which is crucial to their interpretation in Fig 2a and c, is not related with the image-processing procedure.

We thank the reviewer for pointing out this ambiguity, which we have now treated using Figure S3 in the supporting information. As shown in this figure, the unprocessed data exhibits the same registry shift as the averaged derivative image, confirming that our processing steps only highlight the atomic structure, but do not affect any relative lattice positions. We also confirmed that no shifts in the relative position of basal and edge protrusions occur in the simulated STM images upon taking their derivative.

Finally, for a more complete description of the image processing procedure, we now note in the caption of Figure 2 that the derivation was performed in the fast scanning direction.

Reviewer #2 (Remarks to the Author):

After a considerable long-time modification, the quality of the manuscript was greatly improved. In particular, all my raised concerns were positively responded, therefore, I’d like to recommend its publication in Nat. Communications.

REVIEWERS' COMMENTS:

Reviewer #1 (Remarks to the Author):

The authors have now revised the manuscript and have fully addressed the issues related with the comparison and experiments. The authors have added new figures to show how the grids are placed on the images and revised the text accordingly.

In conclusion, I recommend that the paper can be accepted for publication.